# A qualitative exploration of the impact of the COVID-19 pandemic on gender-based violence against women living with HIV or tuberculosis in Timor Leste

**Nelson Martins[1,2,3◐], Domingos Soares[4,5◐], Caetano Gusmao[4‡], Maria Nunes[1], Laura Abrantes[1], Diana Valadares[1], Suzi Marcal[1], Marcelo Mali[6], Luis Alves[1], Jorge Martins[1], Valente da Silva[1,2‡], Paul Russell Ward[7], Nelsensius Klau Fauk[7]***

1 Daslo Research and Development, Timor Leste, 2 Universidade da Paz (UNPAZ), Timor Leste, 3 Menzies School of Health Research, Darwin, NT, Australia, 4 Instituto Nacional de Saúde Publica Timor-Leste (INSP-TL), Ministry of Health Timor-Leste, 5 Instituto Superior Cristal, Timor Leste, 6 Ministerio da Saúde (MdS) Timor-Leste, 7 Centre for Public Health, Equity and Human Flourishing, Torrens University Australia, Melbourne, Australia

◐ These authors contributed equally to this work.
‡ These authors also contributed equally to this work
* nelsensius.fauk@torrens.edu.au

**Data Availability Statement:** All data are in the paper and/or its Supporting Information files.

## Abstract

Violence against women or gender-based violence (GBV) is a significant public health issue facing women and girls in different settings. It is reported to have worsened globally during the COVID-19 pandemic. Despite the impact of the COVID-19 pandemic on increased violence against women in general, which has been reported in many settings globally, there is a paucity of evidence of its impact on violence against highly vulnerable women living with HIV or tuberculosis (TB). Using a qualitative design, this study aimed to explore the views and experiences of women living with HIV (n = 19) or TB (n = 23) in Timor Leste regarding the GBV they faced during the COVID-19 pandemic. They were recruited using the snowballing sampling technique. Data were collected using one-on-one, in-depth interviews and focus group discussions. The five steps of qualitative data analysis suggested in Ritchie and Spencer's analysis framework were employed to guide the analysis of the findings. Findings indicated that women in this study experienced intensified physical, verbal, sexual and psychological violence by their partners, spouses, in-laws, and parents or other family members during the COVID-19 pandemic. Several prominent risk factors that worsened violence against women during the pandemic were (i) HIV or TB-positive status, (ii) traditional gender roles or responsibilities and expectations, (iii) economic and financial difficulties reflected in the loss of jobs and incomes due to the pandemic, and (iv) individual factors such as jealousy and increased alcohol drinking developed during the lockdowns. The women's experience of GBV during the pandemic also led to various negative psychological impacts. The findings underscore the urgent need for multifaceted interventions to address GBV, which should encompass challenging traditional gender norms, addressing economic inequalities, and targeting individual-level risk factors. The findings also indicate the need for the development of robust monitoring and evaluation systems to assess the effectiveness of policies

**Funding:** Global Fund Timor Leste (East Timor): No. Ref.387/DFG. The funders had no role in study design, data collection and analysis, decision to publish, or preparation of the manuscript.

**Competing interests:** The authors have declared that no competing interests exist.

and interventions addressing GBV where the results can inform future improvement. The findings also indicate the need to include GBV in the protocol or guidelines for HIV and TB management. Future large-scale quantitative studies to capture the magnitude and specific drivers of GBV against women living with HIV and TB during the pandemic are recommended.

## Introduction

Violence against women, commonly known as Gender-Based Violence (GBV), is a global problem. The United Nations defines violence against women as "any act of gender-based violence that results in, or is likely to result in, physical, sexual, or mental harm or suffering to women, including threats of such acts, coercion or arbitrary deprivation of liberty, whether occurring in public or in private life" [1]. It is regarded as a devastating human rights violation and a major impediment to attaining sustainable development goals on gender equality in 2030 [2]. Globally, about one in three women (30%) have been subjected to either physical or sexual intimate partner violence (IPV) or non-partner sexual violence in their lifetime [2,3]. Power imbalance, culture, internal conflicts, war, and displacement have been reported as common risk factors for violence against girls and women by men [3]. Natural disasters and disease pandemics that cause humanitarian crises are also factors that increase violence against women and girls by men [4–6]. This often stems from increased mental health issues such as trauma and stress due to the disaster or disease pandemic and is triggered by the difficult situations facing them, financial concerns, inability to fulfil necessities, and poor living conditions during or after the disaster or disease pandemic [4,6,7].

The recent COVID-19 pandemic has also been reported as a significant contributing factor that has worsened GBV against women globally, both in developed and developing countries, by spouses and partners or former partners [3,8–10]. Despite these previous findings, in this study, we explored GBV experiences of specific and highly vulnerable groups of East Timorese women living with HIV or tuberculosis (TB) that have not been covered in previous studies in other settings [8,11]. Intensified violence against women in general in different settings during social distancing periods and lockdowns can be categorised into physical, sexual, and psychological or emotional violence. Physical violence, reflected in physical assaults such as hand smashes and kicking, was commonly experienced by women during the COVID-19 pandemic or the implementation of the COVID-19 lockdowns in several countries, such as Singapore, Canada, France, Australia, Iran, and Ethiopia [11–16]. Other instances of physical violence against women in Australia during the pandemic included men restricting women's movement, keeping women isolated, gaining access to women's residences and coercing women into residing with them using COVID-19 restrictions [10,16]. Women in some countries like Italy, Singapore, Australia, France, Canada, Iran and Ethiopia are also reported to experience increased sexual violence or abuse committed by their spouses or partners, which was reflected in pressured sex and coercive sex during the COVID-19 lockdowns [10,12,14,15,17,18]. Psychological or emotional violence or abuse was also commonly experienced by women and girls in countries such as France, Australia, Iran, and Ethiopia during the social distancing period and lockdowns, where they had to spend more time at home with and were controlled by male perpetrators [10,12,13].

Timor-Leste seems to have significant rates of GBV, with limited evidence suggesting that approximately one-third of Timorese women have encountered GBV, which includes physical

abuse, coercion, harassment, or being deprived of freedom, and around 59% of Timorese women aged 15–19 have undergone sexual violence at some point in their lives [19,20]. With the first HIV case reported in 2003, Timor-Leste has registered 1,704 cases, and a significant increase in HIV prevalence has been reported among pregnant women and key population groups [19,20]. Women living with HIV, along with other vulnerable groups, face challenges in negotiating safer sex and accessing prevention resources [21]. Compounding this issue is the prevalent stigma and discrimination against people living with HIV/AIDS, impacting their lives and access to healthcare [22]. Despite a low overall prevalence of 0.30%, there is a concerning trend toward a broader HIV/AIDS epidemic, extending beyond key populations to the general population [19,23]. In addition, Timor Leste has the second-highest TB incidence rate in Southeast Asia, 486/10000 population in 2022 [24]. Factors contributing to TB infections include smoking, alcohol consumption, malnutrition, and poverty [24].

One of the main risk factors for GBV facing women and girls during the COVID-19 lockdowns was unwanted sexual contact or women refusing sexual intimacy with their spouses or partners [12,14,15]. This was reported to trigger anger and spousal disputes, which led to physical, sexual, and emotional violence against women by their male partners or spouses. Economic insecurity reflected in financial stress due to reduced or loss of income experienced by households, male partners or spouses, and the unavailability of job opportunities during the COVID-19 lockdown also caused stress and heightened pressures on men [8,11,25,26]. Such a situation and its psychological consequences were identified as risk factors for violence against women by their partners or spouses [8,11,25,26]. Such a situation was reported to be even more challenging for women within families where men are the primary income earners [8]. Increased alcohol consumption and illicit drug use among male partners or spouses during social distancing periods and lockdowns, which hindered constructive spousal conversations, were also reported as risk factors for physical assaults and sexual violence against women during the pandemic [8,11,25]. Violence in any form experienced by women during the pandemic was reported to be exacerbated by other factors such as women's inability to seek or access support services due to being restricted by male perpetrators, aggressive behaviours of male partners or spouses, decreased protection and social support and limited or unavailability of facilities providing support [8,16,26]. Being younger and having more children were also significantly associated with GBV against women during the COVID-19 lockdown [13].

There have been associations between the COVID-19 pandemic and violence against women in general in many settings globally [8,11] and predictions about the possibility of the increase of GBV against women living with HIV during the COVID-19 lockdowns and social distancing [27,28]. Such predictions are supported by the global evidence suggesting a high prevalence of women living with HIV encountering some form of violence during their lives [29]. Additionally, women with HIV face a disproportionate burden of various inequalities, such as elevated rates of GBV, economic difficulties, lack of access to food, and precarious housing situations [27,28]. Consequently, there is a specific worry that mandates to 'stay at home' may be unfeasible and hazardous for numerous women living with HIV [27,28]. Findings from two studies in Brazil and the US suggest the prevalence of 27.8% and 79.6% GBV against women living with HIV in these countries, respectively, during the pandemic [30,31]. Unemployment status, being diagnosed with post-traumatic stress disorder, the use of illicit drugs, binge drinking, and loneliness have been reported as the associated factors for GBV against them [30,31]. However, there is a paucity of evidence on the intersections of COVID-19, GBV, HIV, and TB in women or the impact of the COVID-19 pandemic on GBV against women living with HIV or TB and its risk factors in many other settings globally, including in Timor-Leste. Women living with HIV or TB are highly vulnerable groups carrying a double burden of their HIV or TB-positive status that often leads to them being labelled negatively,

treated discriminatively [32–34], and their gender as women who are often socially and cultur-ally subordinated to men within societies in developing countries, including Timor Leste [35–37]. Understanding GBV experiences facing these vulnerable women during the COVID-19 pandemic and its risk factors is important as this can help inform the development of policies and interventions that address their specific needs and support them in taking appropriate action towards violence. This study aims to fill these gaps in knowledge by exploring in-depth the views and experiences of women living with HIV or TB regarding GBV and its risk factors during the COVID-19 pandemic in Timor Leste.

## Methods

### Study settings

This study was conducted in Timor Leste, a small half-island country of 15,007 km$^2$ divided into 13 municipalities, one autonomous region, 67 administrative posts and 452 villages [38]. It has a population of 1,340,434 [38]. Health has been a priority since Timor Leste gained inde-pendence, as demonstrated in its Constitution, where Article 57 guarantees the fundamental right of each citizen to access free health care [39]. Health Indicators have improved substan-tially since 2002, as reported in the Timor Leste Demographic Health Survey (TLDHS) 2016 [20]. Timor Leste is a country with significant rates of GBV, with one in three Timorese women having experienced domestic violence [20]. Regarding the HIV epidemic, although the prevalence is low at 0.30%, evidence suggests an increasing trend of the epidemic in Timor Leste, with the majority of HIV cases being diagnosed among women aged 15 and above [19,40]. With regards to TB, the incidence rate in Timor Leste is reported to be 486 per 100,000, which places Timor Leste at the second-highest TB incidence rate in the Southeast Asia region [24].

### Study design

This is a qualitative study employing a phenomenological approach to understanding the phe-nomenon of GBV or violence against women living with HIV and TB during the COVID-19 pandemic in Timor Leste. The qualitative design was applied as it is considered appropriate and effective when exploring participants' views and real-life experiences [41,42]. It enabled the exploration of the women's stories, understandings, and interpretations of GBV they faced during the COVID-19 pandemic [41]. It also facilitated the researchers' exploration and understanding of the values and meanings the women had regarding GBV facing them in their daily lives [41].

### Participant recruitment and data collection

Participants in this study were women living with HIV or TB in eight municipalities (Ainaro, Bobonaro, Covalima, Ermera, Liquiça, Manufahi, Region of Oé-Cusse Ambeno and Dili) in Timor Leste as requested by the Ministry of Health (MOH) and the Global Fund (GF) Timor Leste, that commissioned this project. Purposive and snowballing sampling techniques were used for participant recruitment. Before the recruitment, the Chief of each municipality was met to explain the study and obtain a permission letter. The researchers purposively approached CHCs in each municipality and asked permission to distribute the study informa-tion sheets to potential participants through the information boards and front service desks of the TB and HIV units. Potential participants who contacted the field researchers and con-firmed their participation were recruited for interviews. This was then followed by the snow-balling sampling technique, through which the initial participants were asked to help

distribute the study information sheets to their eligible friends and families. They were from different community groups in the study settings. The researchers and participants were unknown to each other prior to the study. Participants were recruited based on several inclusion criteria, including one aged 19 years or above, living with HIV or TB, and willing to participate in the study voluntarily. Finally, 42 women (19 women living with HIV and 23 women living with TB) participated in this study, with 24 participating in in-depth interviews and 18 participating in focus group discussions (FGDs).

Data was collected using one-on-one in-depth interviews and FGDs from 21 October 2022 to 28 February 2023. Interviews were conducted in a private room at the CHCs and at a time mutually agreed upon by the researchers (MN, MAM, VS) and each participant. FDGs were conducted in a private room at the CHCs (MN, MAM, VS). Interviews and FDGs were conducted in Tetum, the national language of TL, which both the field researchers and participants fluently speak. Interviews and FDGs were digitally audio recorded, and field notes were taken during the interview and discussion. Interviews and FDGs focused on several key areas, such as the participant's perceptions and experiences of physical, verbal, psychological and sexual violence within their families, relationships, and communities; participant perceptions of factors that contributed to violence against them within these settings. The details of the research questions are provided in supplementary file one (SF 1). The duration for the interviews and FDGs was approximately 45 to 50 minutes. The recruitment of the participants and interviews ceased once the researchers felt the data had been rich enough to address the research questions and objectives. The richness of the data was determined based on data saturation, which was indicated through the similarity of responses provided by the last few participants to those of previous participants [43]. No repeated interviews were conducted with any participants, and no established relationship between the researchers and the participants existed before the study. Considering the sensitivity of the collected information and to prevent the possibility of negative consequences towards the participants, we decided not to return the transcripts of the interviews to each participant to review before we performed the analysis.

## Data analysis

Before the comprehensive data analysis, the audio recordings of the interviews and FDGs were transcribed manually and verbatim (MN, MAM, VS), and fieldnotes were integrated into each transcript. The transcripts were then imported to NVIVO version 12.1 pro, 2020, for comprehensive data analysis. Data analysis was guided by a qualitative data analysis framework that suggests five steps for managing qualitative data in a coherent and structured way and guided the analytic process in a rigorous, transparent and valid way [44,45]. Firstly, familiarisation with the transcripts through repeated reading of the transcripts. During the reading process, information in each transcript was broken down into small pieces of data extracts, and comments and labels were given to each data extract; secondly, the identification of a thematic framework was performed by listing recurrently emerging key issues and concepts which were then used to develop the thematic framework; thirdly, indexing the entire data starting with creating a long list of open codes to identify similar and redundant codes or nodes and collated them to reduce the number. This was followed by closed coding, where codes or nodes that refer to the same theme or sub-theme were grouped. For example, a range of codes regarding the participants' (both married and non-married) various experiences of GBV from spouses, family members, and in-laws, including physical (e.g., being beaten, kicked, or slapped), sexual (e.g., forced sexual intercourse), emotional (e.g., being pressured), and verbal (e.g., insults, being scolded or shouted at) abuses were grouped under the theme: 'women's experiences of GBV during COVID-19 pandemic'. Similarly, various codes on how HIV or TB-positive status

led to physical and verbal violence against women, and supporting conditions, such as living with other family members in the same house, the dislike of others towards the women's HIV or TB-positive status were grouped under sub-theme: "*HIV or TB-positive status of women*". Codes on women's responsibilities in the family and other family members' expectations of them, the position of women or wives in the family and the supporting conditions, such as weak physical condition, feeling tired, and staying at home for an extended period during lockdowns, were classified under the sub-theme: "Women's responsibility within households". Codes regarding economic and financial difficulties experienced during the COVID-19 pandemic, loss of job, and financial incapability to pay their children's tuition fees and fulfil necessities, and how these led to women being divorced by their husbands, the sale of properties or personal belongings and spousal disputes, were placed under the sub-theme: "Economic and financial difficulties during COVID-19 pandemic". The three sub-themes were used to form the theme: "Risk factors for GBV against women living with HIV or TB". Likewise, various codes explaining individual attitudes and behaviours of male partners, spouses or other family members, such as disapproval of parents, spouses, or other family members about women being late returning home after activities, partners' or spouses' jealousy and excessive alcohol drinking, and how these led to spousal dispute, physical, verbal, and psychological violence against women, were grouped under the theme: "Individual-level risk factors for GBV against women during the COVID-19 pandemic". Finally, codes on how GBV led to psychological impacts on the women, such as emotional repression, heightened stress levels, fear, and feelings of loneliness and isolation, were grouped under the theme: "Psychological impacts of GBV on women living with HIV or TB"; fourthly, charting the data by reorganising each theme and its codes or nodes in a summary of chart for comparisons within each transcript or across transcripts; and finally, mapping and interpretations of the entire data [42,46].

Data analysis and coding were conducted in Tetum language to retain the cultural, religious and social meanings attached to the information provided by the women [47]. The selected quotes for this publication were translated into English by NM, MAM, and NKF, who are fluent in Tetum and English. To maintain the accuracy of the translation and the interpretations, other authors, who speak both languages, checked and re-checked both versions [48]. Although data analysis was carried out by NM, MAM, and NKF, the findings were presented to the team, and team-based analysis and discussion were performed at regular meetings. Team discussions were held throughout the data analysis process to maintain the reliability and validity of the data, and all authors agreed upon the final themes and interpretations reported in this paper.

## Ethical consideration

The study obtained ethics approval from the National Health Institute- Health Research Ethics Committee, MOH-Timor Leste (No: 229/MS-INS/GDE/X/2022). During the recruitment and before commencing the interviews and FDGs, participants were informed about the aim of the study and the voluntary nature of their participation. They were also advised that there would be no consequences if they could withdraw from the study without giving any reason. They were assured that the anonymity and confidentiality of the data or information provided in the interviews and FDGs would be maintained to prevent the possibility of linking the information to any individual in the future. This is done by giving specific letters and words to each transcript and quote. Before commencing the interviews, the participants were given a chance to read a written informed consent and sign it. For the participants who could not read, the field researchers read the content of the informed consent for them. All participants were given a chance to raise questions or ask for clarification after reading or listening to the informed

consent. They were also informed during the recruitment that they were allowed to bring somebody they trusted to be with them during the interviews or FGDs, but none of them did so. All participants signed and returned a written informed consent to the field researchers on the interview day.

## Results

### Sociodemographic profile of the participants

A total of 42 women living with HIV or TB participated in this study. They were in the age range between 19 and 70 years (see Table 1). Most participants were married, while four were divorced or widowed, and another four were never married. Most of them never attended formal school (illiterate) or graduated from elementary school. Some graduated high school, and only three had bachelor's degrees. Most women were housewives or single mothers with no jobs.

### Women's experience of GBV during COVID-19 pandemic

This section specifically addresses the study objective of exploring the women's experiences with GBV in different manifestations amid the COVID-19 pandemic, encompassing instances of physical, sexual, verbal, and emotional abuses. Violence against women by a male partner or spouse was experienced by both participants living with HIV and TB, especially during the COVID-19 pandemic. Their stories illustrated that GBV against them, whether in the form of physical, sexual, or emotional violence, worsened during the pandemic. They were often the target of their partner's or spouse's anger, which usually resulted in the use of harsh words

**Table 1. Sociodemographic profile of the participants.**

| Characteristics | Women Living with HIV and TB (N = 42) |
|---|---|
| **Age** | |
| 19 | 1 |
| 20–29 | 9 |
| 30–39 | 15 |
| 40–49 | 8 |
| 50–59 | 6 |
| ≥60 | 3 |
| **Level of Education** | |
| Bachelor | 2 |
| Senior high school | 11 |
| Junior high school | 2 |
| Elementary school | 12 |
| Never attended school | 15 |
| **Marital status** | |
| Married | 33 |
| Divorce | 4 |
| Widow | 1 |
| Never married | 4 |
| **Occupation** | |
| Housewife (unemployed) | 37 |
| Retired | 2 |
| University student | 1 |
| Volunteer counsellor | 2 |

against them, physical beatings and psychological pressure that made them feel stressed, afraid, and anxious, and experience prolonged physical and emotional pain. The following narratives from two married women reflect the above situations experienced by the women who participated in this research:

> "My problem at home is that, during the pandemic, my husband beat me every day, kicked my back, punched, slapped. That's the problem; he got mad quickly; he did not love me and wanted me to die, cursed me to die, that's why I got sick like this, . . .. he hit me on my back, that's why I got sick like this, we argued, we fought, he hit me with his hand" (FGD, participant with TB).

> "When my husband was angry, he would beat and swear at me. COVID-19 caused us a lot of problems" (FGD, participant with HIV).

Similar experiences were also expressed by other married women living with HIV or TB in their stories about the acts of sexual violence from their partners or spouses during the COVID-19 pandemic. Having no sexual desire and being forced to engage in sexual intercourse were conditions reflecting their experience of sexual violence in their spousal relationships during the pandemic. Male partners or spouses spending more time at home during the lockdowns and the women's inability to reject forced sex were mentioned as the supporting factors for such violence against them. The following narratives illustrate that married women were not only being forced to have unwanted sex but were in a powerless position, making them unable to refuse sexual demands from their partners or spouses:

> ". . .. my husband often forced me during the pandemic as he was at home a lot. I didn't like being forced. If I have [sexual] desire, then I can do it; if not, then I don't want. But as a wife, I can do nothing. . . .. So if my husband forces me to have sex, I have to accept it" (Interview, participant with TB).

The married women's experience of violence during the COVID-19 pandemic did not only come from their partners or spouses but also from their in-laws. Several married women revealed the experience of verbal abuse and rejection from their spouse's family, such as being scolded or shouted at daily by in-laws. Living with in-laws in the same house, which is a common practice and socially and culturally acceptable in TL, and their inability to work during the pandemic were some of the main supporting reasons for the abusive attitudes and behaviours from their in-laws. At the same time, a situation of living together with in-laws seemed to force them into facing such abuses and restrict their movement to avoid meeting in-laws:

> "We [the woman and her husband and kids] live together with in-laws. During the pandemic, we were often at home together, and I was sick; my mother-in-law always yelled at me, saying that because of my sickness, I didn't want to work. . . .. I was feeling the rejection from her and others" (FGD, participant with TB).

Experiences of violence during the COVID-19 pandemic were also reflected in the stories of unmarried participants across the study settings. Unlike the married ones, the violence facing them came from their family members (e.g., parents, siblings, and relatives) and neighbours. The interviews revealed that their experiences of violence from their families during the COVID-19 pandemic were more in the form of arbitrary deprivation of their liberty that caused mental harm or suffering to them or negatively influenced their mental condition. Meanwhile, some were also insulted by neighbours through cynical questions, which caused

them a big concern and worry about being avoided or losing friends due to the disease they had. The following participants who had been living with HIV and TB for several years reflected such experiences in their narratives:

> *"During COVID-19, my family isolated me at home and did not let me out in the community or contact with other people . . .. at times, my colleagues asked me to participate in some activities I couldn't participate, and I felt stressed, or a lot of thinking started appearing in my mind (thinking a lot)" (FGD, participant with TB).*

> *"During the COVID pandemic, I felt sad and worried about my neighbours because they sometimes insulted me by asking me how come my family and I had TB and HIV. It was a big concern for me because I am a [university] student, and I am still afraid that my friends will stay away from me and hate me due to the disease that I have." (Interview, participant with HIV).*

The latter participant further explained that she had not encountered such a cynical and harsh question before the pandemic because she used to attend classes on campus daily and didn't frequently interact with neighbours: "*Before COVID, I seldom conversed with them [neighbours] because I spent most weekdays on campus attending classes.*" She highlighted that neighbours within her communities interacted more often during the lockdowns: "*It seemed that due to the lockdowns, neighbours connected or encountered each other more frequently than before COVID.*"

## Risk factors for GBV against women living with HIV or TB

This theme addresses the research objective of understanding risk factors for GBV against women during the COVID-19 pandemic. It outlines several main risk factors, such as the HIV or TB-positive status of women, their responsibility within households, economic or financial difficulties, and a spectrum of individual-level risk factors.

**HIV or TB-positive status of women.** Having HIV or TB-positive status in Timor Leste not only caused negative impacts such as stigma and discrimination against women within the families and communities but also GBV against women by other family members. Physical violence, such as being kicked, slapped, pushed, and chased away from home, and verbal violence, such as being yelled at and shouted at by other family members, were examples of violence experienced by participants due to their HIV or TB-positive status. Several women explained further that the physical and verbal violence they experienced due to living with HIV or TB increased during the COVID-19 pandemic. Some of the underlying reasons for such violence against these women included living with other family members in the same house or seeing each other daily during the pandemic and the dislike or disgust of other family members towards the women's HIV or TB-positive status:

> *"I got so many insults from my brother and sister-in-law. They kicked me out of home. They told me that I had a bad illness and that I should stay away from them. . . ... They often yelled and screamed at me when I stayed close to them. During the COVID-19 lockdowns, we were at home most of the time; I couldn't stay away from them because we lived in the same house. That is why the insults I got from them intensified during those lockdowns. . . .. They talked badly and swore me many times" (Interview, participant with HIV).*

"But I felt sad and worried about my neighbours because they sometimes insulted me. I got physical violence from family or neighbours, such as insulting. . . .. Sometimes, my family were angry with me and said some words that made me hurt. When I was alone, I often asked myself, I don't want to have this disease [TB], but why are they angry with me? They

hate me because I'm infected. I got verbal violence from my family, and my friends sometimes said some words that made me feel hurt" (FGD, participant with TB).

*"There is no love and peace in my family. They don't like me being at home. During COVID, I went from one house to another. When I got home, my mom would ask me to go out and say I was a foreigner and shouldn't stay at home. When I went to my brother's home, they kicked me out and said they didn't want me to visit their home because I have the disease [HIV]. (FGD, participants with HIV).*

Some other participants also expressed similar experiences of violence by neighbours or members of the community where they lived due to their HIV or TB-positive status, as illustrated in the following narratives:

*"It's true; during the COVID-19 lockdowns, some of my neighbours talked negatively about me. I heard words that made me feel hurt by my neighbours. They talked about my disease. They said I was now sick, and they shouldn't be close to me; otherwise, it would spread to them. Even though they did not speak directly to me, I heard from some neighbours who cared for and were close to me, and when I heard this, I felt hurt" (Interview, participant with HIV).*

*"Some neighbours said bad things about me because I have this disease. Although they didn't abuse me physically, I felt the pain when I heard what they talked negatively about me" (Interview, participant with TB).*

**Women's responsibility within households.** Women's responsibilities in the family and other family members' expectations of them were triggering factors for verbal and psychological violence against them by other family members once they failed to perform during the COVID-19 pandemic lockdowns. Within society in TL, women, especially the married ones, are expected to be responsible for household chores, taking care of their children's and spouses' needs within their households, and earning some income. Failing to perform their responsibilities may lead to being scolded, accused, and expelled from home by other family members, as experienced by several married participants during the COVID-19 pandemic. Weak physical condition and feeling tired due to HIV or TB infections were the underlying reasons for their failure to perform household chores, leading to violence against them during the pandemic. Likewise, staying at home for an extended period during lockdowns was also brought up as a triggering factor for complaints against them by other family members, even though these women were physically ill and unable to perform their household chores optimally:

*"They [other family members] don't understand my condition. I have been at home most of the time during the COVID-19 pandemic and lockdowns, so once they saw that I didn't do my responsibilities, they didn't accept it and scolded me. They expected me to do everything as normal and were not aware that I was so sick, I wasn't strong physically [participant looked sad and cried]" (Interview, participant with TB).*

*"Yes, I often got scolded, yelled at, and screamed at during the lockdowns. They [her family members] even told me to leave home and often blamed me even when I was doing housework. I was so sick and very weak due to this infection, but they kept on asking me to work, saying if I didn't work, I would not eat" (FGD, participant with HIV).*

The same experience was also described by other married women who lived in the same house with their in-laws. The in-laws' expectations of them were not only to do household chores but also to earn money to support their own family and children. Such expectations seemed to be risk factors for verbal and psychological violence these women experienced from their in-laws. Interviews with these women uncovered how they were scolded with harsh words and told to work by in-laws, regardless of their poor health condition and difficulty finding a job due to a lack of job opportunities during the COVID-19 pandemic. The following stories of these married women depict how their in-laws had no sympathy towards their health condition and the use of abusive words against them by in-laws was described to exacerbate their psychological state:

> "We [the woman and her husband and kids] lived at the same house as my in-laws. I was so sick during the pandemic, but my mother-in-law didn't care. I didn't work because I was sick, but she kept yelling at me and pushing me to work even though I didn't want to because I easily got tired and was very weak. She said there is medication I could take and then work. If I don't work, then who will feed my family and my kids" (FGD, participant with TB).

> "We lived with my parent-in-law. When I was sick, my mother-in-law screamed, asking me to find a job and work" (Interview, participant with HIV).

The position of women or wives in the family was also one of the factors that contributed to GBV against the participants within families across the study settings. In Timorese culture, husbands are regarded as the head of the household and the ones making decisions for the family and the women or wives are supposed to listen to the husbands. Some women's narratives reflected such cultural practice, which seemed to influence them, positioning themselves as inferior to their husbands and obliging them to submit to the head of the household despite experiencing abuse or violence during the COVID-19 pandemic:

> "In our culture, when our husband speaks, the wife should be quiet. Women don't talk back [not to talk much]. Women should know this cultural practice, so women don't talk when the men are talking. There were many times during COVID where he [her husband] scolded me or was being rude in his words or behaviours, but I couldn't go against him every time we have a problem" (FGD, participant with HIV).

> "It's normal for a married woman not to be carelessly out. I also have a lot of work to do at home, such as cooking and washing clothes. So, it was correct that my husband didn't allow me to go out during the pandemic even though it felt too much . . .." (Interview, participant with TB).

> "I was just quiet and didn't make any problem, especially during the pandemic because I knew that I could get beaten by my husband. It happened before, and I didn't want it to happen again" (FGD, participant with HIV).

**Economic and financial difficulties during COVID-19 pandemic.** The economic and financial difficulties experienced during the COVID-19 pandemic were also one of the risk factors for violence against women participating in this study. The economic challenges were reflected in the loss of jobs and incomes experienced by women and their partners or spouses during the pandemic. For some women, such a situation led to financial difficulties, which caused stress, worry, anger, and dispute between them and their spouses, leading to physical

and emotional violence against them. Loss of job during the COVID-19 pandemic and lockdowns provided couples with more time to spend together at home, which for some participants in this study was another risk factor for violent attitudes and behaviours of their spouses against them. The following narrative of a woman living with TB reflects such situations she had gone through in her family during the pandemic, which led to violence against her:

> "My husband worked at a store in Dili [capital city of Timor Leste], but because of the COVID-19 outbreak, my husband lost this job and only stayed at home. I was not working either, just looking after my children. We had no money, . . .. When my husband was angry, he would beat and swear at me. He abused me. I think he did that because he was stressed and felt pressured. Because COVID-19 came, our products were not sold, and there was also no money, which caused us problems. I was stressed because of no money, my ill-health condition and his violent attitudes and behaviours" (FGD, participant with TB).

The above women's narrative showed that she had not encountered such experience prior to the pandemic: "*He [her husband] wasn't used to being angry like this before COVID. Besides, we only saw each other on weekends because he worked in the city.*" The economic and financial difficulties experienced during the COVID-19 pandemic also negatively influenced the women's financial capability to pay their children's tuition fees. Several women described that such difficulties and the inability to afford the education fees of their children often led to verbal abuse against them by their spouses, which negatively impacted their mental health and wellbeing during the pandemic. For some women, the economic and financial difficulties led to the withdrawal of children from school, which also made them sad and stressed, as reflected in the following quote:

> "I want to share the responsibility for looking after my children's school and supporting their studies. When I told him [her husband] about the children's tuition fees, he answered with anger, swearing at me and told me to stop sending our children to school. I'm very sad. For me, school is very important. I didn't go to school, so I want my children to go to school" (FGD, participant with TB).

> "During COVID-19, I was in the countryside with children, and my husband never sent us money. That made me think a lot. I felt so stressed. I was abused if I asked for money. . .." (Interview, participant with HIV).

Furthermore, the economic and financial difficulties experienced by the women and their families during the COVID-19 pandemic not only caused negative consequences in the form of violence against women in the household but also led to women being divorced by their husbands. Moreover, several women commented that financial difficulties they faced during the pandemic due to the loss of their job led to the inability to fulfil necessities, the sale of properties or personal belongings and spousal disputes or physical violence against them:

> "My husband is the one who was beating me because he lost his job during the pandemic, and I didn't have any job. We were married at a young age . . .. We don't have money, we often have conflict, and he often beat me up during the pandemic" (Interview, participant with HIV).

> "My husband divorced me during the pandemic when we were in a tough situation financially, and I felt sad. I'm depressed. My children keep asking me where their dad is. We don't have money, and there is no money to buy rice. During COVID-19, we didn't have money,

*and there was no food. I sold out all my gold necklaces to get money to buy rice" (FGD, participant with TB).*

*"We had problems and conflict, and then we divorced. We divorced during the COVID-19 pandemic (Interview, participant with HIV).*

**Individual-level risk factors for GBV against women during the COVID-19 pandemic.** Individual attitudes and behaviours of male partners, spouses or other family members were also factors that contributed to GBV against the participants during the pandemic. For example, the disapproval of parents, spouses, or other family members when they were late returning home after carrying out activities elsewhere was a supporting factor for verbal and psychological violence against several women in this study. Some of the reasons for such disapproval of others were the dislike and fear of other family members of COVID-19 transmission:

*"Yes, he [her spouse] threatened me if I came home late from work or after I did some activities outside of our house. He didn't like going out and doing something else" (Interview, participant with HIV).*

*"Sometimes my other family members, especially my parents, disapproved of me going out and coming back home late. They might be afraid of me contracting COVID and transmitting it to them or maybe because I am a woman" (FGD, participant with TB).*

Partners' or spouses' jealousy was another individual risk factor for physical, psychological, and verbal violence against several married women in this study. Their male partners' or spouses' jealous attitude was reflected in the perception that coming back home late after an activity elsewhere was due to the women having an affair with another man. This caused disputes between them, which also led to physical, verbal, and psychological violence against them by their partners or spouses during the pandemic. Some women also commented that the long and repeated periods of lockdown, which resulted in their partners or spouses spending more time at home, seemed to trigger excessive feelings of jealousy, which was not expected before the COVID-19 pandemic:

*"Sometimes he gets jealous when I was hanging out. He didn't allow me to wear short pants. He was suspicious of me having an affair with another man" (FGD, participant with TB).*

Excessive alcohol drinking by the women's partners or spouses was also emphasised as a strong supporting factor for violence against them within families. Several women commented that their spouses were often engaged in excessive alcohol drinking, either alone or with some close friends during the COVID-19 pandemic. This was acknowledged to trigger the men's violent attitudes and behaviours towards them within families:

*"My husband is a drunk man; he provokes me, so I must obey him. He is a madman once he is drunk, and he would beat me up. He did that many times during COVID. If he is not drunk, then he is easy." . . .." (Interview, participant with TB).*

*"Married with a drunk man like my husband, I need to be very patient. He becomes rude and violent once he is drunk. I experienced that during COVID, he drank too much and was violent. So, I should be very patient to deal with him" (FGD, participant with TB).*

## Psychological impacts of GBV on women living with HIV or TB

This section addresses the psychological repercussions of GBV endured by women during the COVID-19 pandemic. These consequences encompassed emotional repression, heightened stress levels, fear, and feelings of loneliness and isolation. Accounts from women depicted experiences of emotional suppression and a sense of isolation and detachment stemming from the GBV they encountered. The majority felt compelled to hide their emotional anguish, striving to maintain an illusion of normalcy to protect themselves and their loved ones from further harm. For instance, one woman living with HIV described feeling stressed and emotionally stifled but chose to remain silent to avoid stirring up trouble, fearing that her actions might lead to her mother receiving abuse from her husband's family. Similarly, another participant shared her reluctance to disclose her emotional turmoil to her family, opting instead to portray a facade of well-being. This suppression of emotions was also echoed by another TB patient who felt obliged to endure her situation silently because of her role as a married woman and an outsider in her husband's community:

> "I was stressed out and felt emotionally suppressed due to the abuses, but I was scared, just quiet and didn't make any problem because my mum could also get beaten by my husband's family. I felt lonely and isolated in this situation" (Interview, participants with HIV).

> "I never tell my family and others about what I have been going through, emotional pain I experience. I pretend to show to them that I'm fine" (Interview, participants with TB).

> "I am a married woman, and I should be patient. And I am only a visitor in my husband's hometown, so I can't go against them. I feel the pressure emotionally due to my husband's harsh attitudes and behaviours, but I couldn't do anything" (Interview, participants with TB).

For some women, the experience of gender-based violence they faced also led to feelings of isolation and alienation due to the inability to confide in others or the absence of supportive individuals. The feelings of loneliness and detachment were stark in the narratives of these women. One woman living with HIV expressed a profound sense of isolation, exacerbated by the absence of someone with whom she could share the emotional burden she faced. This isolation was intensified by the tragic loss of her child, a devastating event met with indifference from her husband's family. Similarly, another woman living with HIV who endured GBV from her husband and in-laws shared a heart-wrenching account of feeling that her world had become empty and dark due to the overwhelming stress and pressure, compounded by a lack of support networks, particularly after the loss of her child. The lack of care and concern from her husband's family during such a traumatic experience accentuated her suffering and deepened her sense of alienation:

> "They [husband's family] insulted me many times and didn't care at all about me and my kids. And even when one of my kids passed away, they didn't come to visit us. My family also never visited me either. I have no one to share the emotional pain I feel" (Interview, participants with HIV).

> "I was insulted, abused by my husband and his family. I felt all the stress and pain. I felt that everything was empty to me and dark, and even more when I lost my child, and a family from my husband's side did not care about me, I was so suffering" (Interview, participants with HIV).

## Discussion

The study explored the impacts of the COVID-19 pandemic on GBV against women living with HIV or TB and its risk factors. The findings of the study highlight the alarming facts of intensified physical, sexual, and psychological or emotional violence against women living with HIV or TB during the COVID-19 pandemic. The experiences of physical violence, verbal abuse and coercive sex by male partners or spouses, arbitrary deprivation of women's liberty and rejection from family members and in-laws shed light on the multifaceted nature of GBV and the impact of the pandemic on the issue among the highly vulnerable women this study. The current findings justify the concerns raised during the early stage of the COVID-19 pandemic about the possibility of its adverse impacts on GBV against women living with HIV [27,28] and enrich the previous findings on the impacts of the pandemic on the increased GBV against women in general globally [3,8–10]. This study also adds to the existing reports of previous studies on various other HIV-related negative experiences facing women living with HIV [33,49,50] and TB [32].

Our study highlights some novel and interconnected findings on the risk factors for GBV against women, particularly during the COVID-19 pandemic. Traditional gender roles and expectations within social and family life are significant aspects illuminated by the current findings as factors that contributed to and heightened GBV against women during the pandemic, which have never been reported previously [8,11,25,26]. These are reflected in the sociocultural perceptions of married and unmarried women as the ones responsible for household chores and childcare. Thus, failure to meet these traditional gender roles and expectations, which intersected with the heightened stressors caused by the COVID-19 pandemic, became a trigger, or created a supporting environment for GBV against women in this study. The expectations of in-laws for married women to generate incomes for supporting their families and children's needs, which seemed to emerge as the result of unconstructive relationships between them, were also another novel finding on risk factors for GBV against married women in this study. The findings indicate the critical need for GBV programs and interventions that consider challenging traditional and social norms and expectations about women to abolish the perpetuation of violence against them and promote gender equality.

Consistent with previous findings on GBV against women in general [8,11,25,26], the current study also highlights economic and financial difficulties as another critical factor that contributed to GBV against both women living with HIV and TB in Timor Leste. Moreover, the current findings provide some detailed mechanisms of how economic and financial difficulties played a role in GBV against women across the study settings. The intensified family financial stress brought about by the loss of jobs experienced by the women's partners or spouses due to the COVID-19 pandemic caused further issues, such as the inability to fulfil necessities, incapability to afford the children's needs and school fees, withdrawal of children from schools, the sale of personal belongings, and spousal disputes within these women's families. These caused stress, worry, sadness and anger for these women and their spouses, which triggered GBV against them. In addition, our findings also strongly indicate that the women's unemployment status and possibly financial dependency on spouses, parents or in-laws placed them even at a lower position and diminished their ability to encounter violence from others within their families. Thus, this study highlights the intersectionality of economic or financial challenges and GVB against women in Timor Leste and suggests the importance of holistic approaches and interventions that address the financial needs of women living with HIV and TB in the country.

The study has also highlighted the significant contribution of individual-level risk factors to the complex dynamics of violence against women living with HIV or TB. For example, male

partners' or spouses' jealous attitudes that stemmed from perceived infidelity, which have never previously been reported, were triggers for GBV against women in this study by their partners or spouses. This is plausible as jealousy, which is 'an emotional response to a threat of losing a valued relationship', often encompasses complex emotions such as anger, suspicion, and humiliation, which can trigger or cause hostile acts of violence against other people [51,52]. Excessive alcohol consumption is another individual-level risk factor for GBV against vulnerable women living with HIV or TB. This supports previous findings suggesting that alcohol consumption is destructive towards constructive conversations in spousal relationships and often becomes the underlying risk factor for violence against women [8,11,25]. The findings have important implications for the development of tailored interventions that address specific attitudes and behaviours that contribute to violence against women.

Furthermore, the current findings highlight the significant psychological consequences of GBV faced by these women, which are consistent with previous findings [53,54]. It underscores a distressing reality facing the women as they struggle to grapple with emotional pressure, heightened stress levels, fear, and feelings of loneliness and isolation because of the abuses, unsupportive surrounding environments, and their inability to confide in others. Such conditions coupled with the fear of further harm towards themselves and their loved ones and additional traumas such as the loss of a child intensified their sense of alienation and exacerbated their emotional suffering. Thus, it is highly likely that the fear of harm could lead to the women's inclination to conceal their emotional distress and prevent them from seeking available support. The findings indicated the importance of comprehensive support systems for these vulnerable women to face the challenges of GBV and its psychological impacts on their lives [55].

Recognising and addressing the gendered impacts of the pandemic in Timor Leste is crucial. The pandemic has exacerbated existing societal inequalities, making women, especially those with HIV or TB, more vulnerable to GBV. Policies in Timor Leste should be specifically designed to support these women, providing not only protection against GBV but also acknowledging the cultural nuances. Challenging and transforming traditional gender roles and expectations, deeply rooted in Timorese society, is vital. These societal norms significantly contribute to GBV, particularly in times of crisis. Interventions aimed at modifying these perceptions and promoting gender equality can create a safer and more equitable environment for women [56].

It is evident that the disease causes financial stress in the family, which is intensified by the pandemic. Disease cause temporal dysfunction in women; it creates dependency, which is a significant factor that increases the vulnerability of women to GBV. Providing financial empowerment and support initiatives can play a crucial role in mitigating these risks and relieving the pressure on the survival of GBV. Additionally, addressing individual-level risk factors, such as jealousy and alcohol consumption among male partners, is essential. Enhancing awareness about GBV in Timor Leste and developing accessible reporting mechanisms, self-reporting, massive media campaigns, and scaling up existing safe spaces for women's survival of GBV are essential. These methods can assist women in safely reporting incidents of GBV [57]. Lastly, strengthening collaboration between the government, NGOs and civil society organisations is critical to developing and implementing comprehensive GBV policies. Such partnerships should ensure dedicated resources and responses specifically tailored to the challenges posed by the pandemic in Timor Leste. This approach can enhance the effectiveness of GBV interventions and provide a framework for ongoing support and protection of women, particularly in the post-pandemic period [56–58].

There is a need to employ mixed-methods research to gain a comprehensive understanding of the issue. Qualitative approaches can provide in-depth insights into the personal experiences of GBV, while quantitative methods can offer broader epidemiological perspectives, in

the other hand exploring the existing policy and strategy implementation will also generate evidence on the policy and strategic gaps.

## Limitations and strengths of the study

The current findings should be interpreted with caution to some possible limitations. The snowballing sampling technique and the distribution of the study information sheets through CHCs might have led to the recruitment of participants from the same networks as the current participants and the recruitment of the participants who had been connected to HIV/TB care and treatment. As a consequence, we may have under-sampled women living with HIV or TB who were not on HIV/TB care and treatment and who may have different stories regarding GBV facing them during the pandemic. We acknowledged that this study qualitative involved small number of participants. As with many other qualitative studies, this study was not designed to claim generalisability of the findings. However, the richness of the findings or information generated from this study can be useful to inform policy and practice. In addition, this is the first study presenting initial in-depth qualitative findings on GBV against women living with HIV or TB in Timor Leste and globally. Therefore, the findings are important to inform policies and interventions to address GBV and its risk factors and impacts on women living with HIV and TB in Timor Leste and other similar setting globally.

## Conclusions

The study presents the perceptions and experiences of women living with HIV and TB in Timor Leste regarding GBV against them during the COVID-19 pandemic. These vulnerable groups of women experienced physical, verbal, sexual and psychological violence by their partners, spouses, in-laws, and parents or other family members during the pandemic. Traditional gender roles and expectations, economic and financial difficulties, and individual factors such as jealousy and alcohol drinking are the prominent risk factors for these types of violence against these women. The findings underscore the urgent need for multifaceted interventions to address GBV, which should encompass challenging traditional gender norms, addressing economic inequalities, and targeting individual-level risk factors. The findings indicate the need for the development of robust monitoring and evaluation systems to assess the effectiveness of policies and interventions addressing GBV where the results can inform future improvement. The findings also indicate the need to include GBV in the protocol or guidelines for HIV and TB management. Future large-scale quantitative studies to capture the magnitude and specific drivers (e.g., socio-cultural and individual factors) of GBV against women and girls living with HIV and TB during the pandemic are recommended.

## Supporting information

**S1 File.**
(DOCX)

## Acknowledgments

We would like to thank the participants who had voluntarily spent their time participating in this study and provided us with valuable information.

## Author Contributions

**Conceptualization:** Nelson Martins, Domingos Soares, Caetano Gusmao, Maria Nunes, Laura Abrantes, Diana Valadares, Suzi Marcal, Marcelo Mali, Luis Alves, Jorge Martins, Valente da Silva, Paul Russell Ward, Nelsensius Klau Fauk.

**Formal analysis:** Nelson Martins, Nelsensius Klau Fauk.

**Funding acquisition:** Nelson Martins.

**Investigation:** Nelson Martins, Marcelo Mali.

**Methodology:** Nelson Martins, Domingos Soares, Caetano Gusmao, Maria Nunes, Laura Abrantes, Diana Valadares, Suzi Marcal, Marcelo Mali, Luis Alves, Jorge Martins, Valente da Silva, Paul Russell Ward, Nelsensius Klau Fauk.

**Project administration:** Nelson Martins, Maria Nunes, Marcelo Mali.

**Writing – original draft:** Nelson Martins, Marcelo Mali, Nelsensius Klau Fauk.

**Writing – review & editing:** Nelson Martins, Domingos Soares, Caetano Gusmao, Maria Nunes, Laura Abrantes, Diana Valadares, Suzi Marcal, Marcelo Mali, Luis Alves, Jorge Martins, Valente da Silva, Paul Russell Ward, Nelsensius Klau Fauk.

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
