## [Decision Letter · Decision Letter 0]

13 Feb 2024

PONE-D-23-41831A qualitative exploration of the impact of the COVID-19 pandemic on gender-based violence against women living with HIV or tuberculosis in Timor LestePLOS ONE

Dear Dr. Fauk,

Thank you for submitting your manuscript to PLOS ONE. After careful consideration, we feel that it has merit but does not fully meet PLOS ONE’s publication criteria as it currently stands. Therefore, we invite you to submit a revised version of the manuscript that addresses the points raised during the review process.

We look forward to receiving your revised manuscript.

Kind regards,

Victor Abiola Adepoju, MBCHB,Msc

Academic Editor

PLOS ONE

Journal Requirements:

"Global Fund Timor Leste (East Timor)"

Additional Editor Comments:

Introduction and BackgroundWeaknesses & Recommendations: •
Lack of Local Context: The introduction mentions global statistics and dynamics of GBV but misses local data critical for contextualizing the study within Timor Leste.•
Actionable Recommendation: The authors should add a paragraph specifically summarizing existing research or reports on GBV, HIV, and TB within Timor Leste. This could be added after the global overview, before transitioning into the study's focus on COVID-19's impact (Introduction, paragraph 3).•
Specific Change: Incorporate statistics from Timor Leste's national health surveys or reports by local NGOs/INGOs that highlight the pre-pandemic situation regarding GBV, HIV, and TB.•
Intersections of COVID-19, GBV, HIV, and TB: The introduction broadly links these issues but lacks depth in literature exploration.•
Actionable Recommendation: Deepen the literature review by including studies that have specifically explored the exacerbation of GBV against women with infectious diseases during crises, with a focus on HIV and TB (Introduction, paragraphs 4-5).•
Specific Change: Add references to studies or theoretical models that discuss the vulnerabilities of women living with HIV or TB during pandemics, detailing how societal, economic, and health system pressures during COVID-19 could worsen these vulnerabilities.

Methods

Weaknesses & Recommendations:

 •
Snowball Sampling Bias: The methodology briefly mentions the use of snowball sampling but does not address potential biases.•
Actionable Recommendation: Explicitly discuss the potential for network bias and over-representation and describe strategies used to ensure a diverse sample, such as reaching out to different community groups or health centers (Methods, Sampling subsection).•
Specific Change: Add a paragraph discussing the limitations of snowball sampling and mitigation strategies employed, such as efforts to reach beyond initial networks.•
Data Analysis Transparency: While the data analysis process is outlined, there are no examples of how themes were developed from data.•
Actionable Recommendation: Provide a detailed example of thematic development, from initial coding to theme generation, possibly in a box or table format to illustrate the process (Methods, Data Analysis subsection).•
Specific Change: Insert a table or text box showing an example of raw data (quotes), initial codes, and how these codes were grouped into themes.

ResultsWeaknesses & Recommendations: •
Link to Research Questions and Objectives: Findings are rich but not explicitly connected back to the study objectives.•
Actionable Recommendation: For each major finding, include a sentence or two that directly ties it back to how it addresses a specific research question or objective outlined in the introduction (Results, start or end of each major finding subsection).•
Specific Change: Add introductory sentences to each subsection of the results that restate the relevant research question or objective the subsection addresses.•
Psychological Impacts Underexplored: There's a missed opportunity for deeper analysis on the psychological impacts of GBV.•
Actionable Recommendation: Include a subsection dedicated to exploring the psychological impacts, supported by direct quotes from participants and a more nuanced interpretation of these experiences (Results, create a new subsection on psychological impacts).•
Specific Change: Expand the analysis of participant narratives to include psychological themes, such as feelings of isolation, stigma, or mental health struggles, with direct quotes to illustrate. Discussion and ConclusionsWeaknesses & Recommendations: •
Policy and Practical Implications: The discussion highlights findings within broader literature but lacks depth on implications.•
Actionable Recommendation: Expand the discussion on policy implications by suggesting specific policy changes or interventions needed in Timor Leste based on the findings, such as integrating GBV prevention strategies into existing HIV/TB programs (Discussion, Implications subsection).•
Specific Change: Add paragraphs that detail specific policy recommendations, including programmatic changes or new interventions targeting the identified risk factors.•
Generic Recommendations for Future Research: The suggestions for future research are broad and not specific.•
Actionable Recommendation: Specify areas where future research is critically needed, perhaps identifying specific populations, geographic areas, or thematic gaps that the current study has uncovered (Discussion, Future Research subsection).•
Specific Change: Provide a list of potential research questions or methodologies that future studies could adopt, based on the gaps identified in the current research.Overall Scientific Soundness and Originality

Recommendations for Improvement:

•
Methodological Rigor: Detail the analytic process more thoroughly to ensure findings are grounded in data.•
Specific Change: Add an appendix or supplementary material that outlines the coding framework or provides examples of the coding process, to enhance methodological transparency.•
Enhance Originality: Draw clearer connections between findings and their implications for Timor Leste's specific context.•
Specific Change: Incorporate a section discussing how the study's findings contribute new insights specific to Timor Leste and how these insights could influence local practice or policy differently from other contexts.

Quality of English Language•
Specific Change: Engage a professional editing service familiar with academic writing in the public health field to ensure clarity and adherence to academic standards. Final RecommendationReconsider after major revision is suggested, with specific emphasis on addressing the methodological and analytical gaps, and articulating the study's contributions more clearly in terms of literature, policy, and practice implications

Reviewers' comments:

Reviewer's Responses to Questions

**Comments to the Author**

1. Is the manuscript technically sound, and do the data support the conclusions?

Reviewer #1: Partly

Reviewer #2: Yes

2. Has the statistical analysis been performed appropriately and rigorously? 

Reviewer #1: N/A

Reviewer #2: Yes

3. Have the authors made all data underlying the findings in their manuscript fully available?

Reviewer #1: No

Reviewer #2: Yes

4. Is the manuscript presented in an intelligible fashion and written in standard English?

Reviewer #1: Yes

Reviewer #2: Yes

5. Review Comments to the Author

Reviewer #1: Please, please include page and line numbering in your submissions. It is almost impossible to indicate where comments should go otherwise!

Introduction

‘using COVID-19 restrictions and the threat of the infection [10, 16].’ What does ‘threat of the infection’ mean? Do you mean they threatened to inform others of the infection?

‘Being younger and having more children and adolescents were also significantly associated with GBV against women during the COVID-19 lockdown [13].’ Do you mean having adolescent children or being an adolescent?

‘Timor Leste’s referral system includes three levels…across the territory [31].’ Is this relevant in an already quite long paper?

Results

Here you mention that many participants were illiterate, but you have not discussed this in the consent or ethics sections. How did you deal with illiteracy when signing consent forms that are by definition written? Did you have witnesses to the explanation of the study? How many participants were illiterate? I would suggest that this might be a key variable in your analysis but does not seem to be covered in the results.

Did women reporting violence also report violent behaviour pre-pandemic? This would be interesting to know. I am struggling to see the connection explicitly (I know, men were at home more etc etc, but I think this needs to be clarified).

For example: ‘During the COVID pandemic, I felt sad and worried about my neighbours because they

sometimes insulted me by asking me how come my family and I had TB and HIV. It was a big

concern for me because I am a [university] student, and I am still afraid that my friends will

stay away from me and hate me due to the disease that I have.”’ Okay, but how is this related to/worsened by Covid?

‘My husband worked at a store in Dili [capital city of Timor Leste], but because of the

COVID-19 outbreak, my husband lost this job and only stayed at home. I was not working

either, just looking after my children. We had no money, …. When my husband was angry, he

would beat and swear at me. He abused me. I think he did that because he was stressed and

felt pressured. Because COVID-19 came, our products were not sold, and there was also no

money, which caused us problems. I was stressed because of no money, my ill-health

condition and his violent attitudes and behaviours” (FGD, participant with TB)’

Again, this quote suggests there was little difference financially pre- and during Covid, except that her husband was at home.

There is some repetition in quotes, e.g.,

‘I got so many insults from my brother and sister-in-law; they kicked me out of the home” (Interview, participant with HIV).’

Next page: ‘I got so many insults from my brother and sister-in-law. They kicked me out of home. They told me that I had a bad illness and that I should stay away from them. ….. They often yelled

and screamed at me when I stayed close to them. During the COVID-19 lockdowns, we were

at home most of the time; I couldn’t stay away from them because we lived in the same house.

That is why the insults I got from them intensified during those lockdowns. …. They talked

badly and swore me many times” (Interview, participant with HIV).’

There is generally quite a lot of repetition in this section, and maybe too many illustrative quotes used to make the same point again.

The individual level risk factors section is probably the most interesting part of the results, but is the shortest. If you have any more material for this section I would include it and cut down some of the earlier sections.

Reviewer #2: I think this manuscript is well researched and written, good job! Minor comments:

1) Interviews and focus groups were conducted in Tetum. Who translated them to English and what was the process?

2) Why did you choose to sample HIV and TB women? What is the motivation? What are their existing risk factors? I know COVID worsened it, especially in the area of GBV. However, were they already experiencing GBV before covid? How does the GBV they experience compare to the rest of the population in TL? This is a missing piece of the puzzle to address. I think it is important to highlight how covid made GBV especially worse for this population, and if it really is the case, why does HIV/TB make life so difficult for these women?

6. PLOS authors have the option to publish the peer review history of their article (what does this mean?). If published, this will include your full peer review and any attached files.

Reviewer #1: No

Reviewer #2: No

---

## [Author Response · Author response to Decision Letter 0]

8 Mar 2024

The responses are attached - see 'Response to Reviewers' file.

---

## [Decision Letter · Decision Letter 1]

11 Jun 2024

A qualitative exploration of the impact of the COVID-19 pandemic on gender-based violence against women living with HIV or tuberculosis in Timor Leste

PONE-D-23-41831R1

Dear Dr. Fauk,

We’re pleased to inform you that your manuscript has been judged scientifically suitable for publication and will be formally accepted for publication once it meets all outstanding technical requirements.

Kind regards,

Laura Kelly

Division Editor

PLOS ONE

Additional Editor Comments (optional):

Reviewers' comments:

Reviewer's Responses to Questions

**Comments to the Author**

1. If the authors have adequately addressed your comments raised in a previous round of review and you feel that this manuscript is now acceptable for publication, you may indicate that here to bypass the “Comments to the Author” section, enter your conflict of interest statement in the “Confidential to Editor” section, and submit your "Accept" recommendation.

Reviewer #1: All comments have been addressed

2. Is the manuscript technically sound, and do the data support the conclusions?

Reviewer #1: Yes

3. Has the statistical analysis been performed appropriately and rigorously? 

Reviewer #1: N/A

4. Have the authors made all data underlying the findings in their manuscript fully available?

Reviewer #1: Yes

5. Is the manuscript presented in an intelligible fashion and written in standard English?

Reviewer #1: Yes

6. Review Comments to the Author

Reviewer #1: (No Response)

7. PLOS authors have the option to publish the peer review history of their article (what does this mean?). If published, this will include your full peer review and any attached files.

Reviewer #1: No

---

## [Editor Report · Acceptance letter]

2 Jul 2024

PONE-D-23-41831R1 

PLOS ONE

Dear Dr. Fauk, 

I'm pleased to inform you that your manuscript has been deemed suitable for publication in PLOS ONE. Congratulations! Your manuscript is now being handed over to our production team.

Kind regards, 

on behalf of

Dr. Laura Hannah Kelly 

Staff Editor

PLOS ONE